# Sounding of sporadic E layers from CSES radio occultation and comparing with ionosonde measurements

Chengkun Gan[1], Jiayu Hu[2] , Xiaomin Luo[3], Chao Xiong[4,5], Shengfeng Gu[1]

[1]GNSS Research Center, Wuhan University, Wuhan 430079, China
[2]School of Geodesy and Geomatics, Wuhan University, Wuhan 430079, China
[3]School of Geography and Information Engineering, China University of Geosciences (Wuhan), Wuhan 430074, China
[4]Department of Space Physics, School of Electronic Information, Wuhan University, Wuhan 430072, China
[5]Hubei Luojia Laboratory, Wuhan 430079, China

*Correspondence to*: Shengfeng Gu (gsf@whu.edu.cn)

**Abstract.** GNSS radio occultation (RO) plays an important role in ionospheric electron density inversion and sounding of sporadic E layers. As China's first electromagnetic satellite, China Seismo Electromagnetic Satellite (CSES) has collected the RO data from both GPS and BDS-2 satellites since March 2018. In this study, we extracted the signal to noise ratio (SNR) data of CSES and calculated the standard deviation of normalized SNR. A new criterion is developed to determine the Es events, that is when the mean value of the absolute value of the difference between the normalized SNR is greater than 3

times of the standard deviation. The statistics show that sporadic E layers have strong seasonal variations with highest occurrence rates in summer season at middle latitudes. It is also found that the occurrence height of Es is mainly located at 90-110 km, and the period of local time 14:00-20:00 is the high incidence period of Es. In addition, the geometric altitudes of a sporadic E layer detected in CSES radio occultation profiles and the virtual heights of a sporadic E layer obtained by the Wuhan Zuoling station (ZLT) ionosonde show three different space-time matching criteria. Our results reveal that there is a

good agreement between both parameters which is reflected in the significant correlation.

## 1. Introduction

The name Sporadic E and its abbreviation Es refer to thin layers of metallic ion plasma which accumulates in the dynamo region of the Earth's ionosphere, mostly between 100 and 125 km, where ion motion is controlled mainly by collisions with the neutrals, thus the ions move with the winds while electrons remain strongly magnetized (Haldoupis 2012). The formation

of sporadic E layer was traditionally attributed to the "windshear theory" (Whitehead 1961; Axford 1963; Whitehead 1989), in which vertical shears in the horizontal wind play a key role in forming these layers from long-lived metallic ions through ion-neutral collisional coupling and geomagnetic Lorentz forcing, vertical shear converge metallic ions into thin sheets of enhanced electron density. More recently, researchers have found multiple factors can contribute to the occurrence of Es, including tidal wind, the Earth's geomagnetic field, and meteoric deposition of metallic material in the background

thermosphere, resulting in variations of Es occurrence with respect to local time, altitude, latitude, longitude, and season

(Haldoupis, 2011; Yeh et al, 2014, Didebulidze et al., 2020). Meanwhile, the ionospheric E region has a relatively higher electrical conductivity and therefore plays a crucial role in the ionosphere electron dynamics at both E- and F-region altitudes (Yue et al. 2015).

Variance in the signal to noise ratio (SNR) caused by strong gradients in the index of refraction has been suggested to identify and sound sporadic E layers (Wu et al., 2005; Arras et al., 2008; Yeh et al., 2012; Hocke et al., 2001; Yue et al., 2015; Tsai et al. 2018). However, in terms of judgment criteria, many scholars propose different selection methods. Chu (2014) set thresholds for signal phase amplitude and carrier phase delay ratio when screening Es, and the ratio of disturbance amplitude to normalized SNR must be greater than 0.01 then it can be counted as Es event. Wu et al. (2005) directly used the

normalized SNR data sequence as the characteristic parameter to detect Es. Arras et al. (2017) and Tsai et al. (2018) used the value of 0.2 as the threshold of the normalized SNR standard deviation sequence. It is considered that Es event occurs when the peak exceeds 0.2. Xue et al. (2018) used 0.1 as the standard deviation threshold to detect single-layer and multi-layer Es events at the same time. Based on GPS radio occultation (RO) techniques, some investigations established global distribution of Es layers information to analyze the climatology of global Es occurrence rates. (Arras et al., 2008; Wickert et al., 2004;

Yeh et al., 2012; Tilo et al., 2014; Arras et al. 2017).

Since the invention of ionosonde in the nineteen-thirties, Es has been investigated extensively from the ground, by means of analyzing ionosonde and incoherent scatter radar observations (Whitehead 1989; Mathews 1998). Ionosondes provide reliable measurements on sporadic E parameters and on the altitude of each layer. The altitudes are given in virtual heights,

and the lower boundary of the sporadic E layer (h'Es). Arras et al. (2017) compared sporadic E altitudes and their intensity with ground based ionosonde data provided by the Digisonde located at Pruhonice close to Prague, Czech Republic (geographic 50°N, 14.5°E) to confirm the derived sporadic E parameters. Wuhan Zuoling station (ZLT) ionosonde (geographic 30.5°N, 114.4°E) is located in central China. It is a representative location due to its low geomagnetic latitude and the longest observational record, which has been well-maintained during the past several decades and its data are of high

quality (Zhou et al. 2021).

China's first electromagnetic satellite, China Seismo Electromagnetic Satellite (CSES), also known as ZH01(01), was successfully launched on Feb 2, 2018. The CSES is a 3-axes-stabilized satellite, based on the Chinese CAST2000 platform, with a mass of about 730 kg and peak power consumption of about 900 W. Scientific data are transmitted in the X-Band at

120 Mbps. The orbit is circular Sun-synchronous, at an altitude of about 507 km, inclination of about 97.4°, descending node at 14:00 LT. All payloads of CSES are designed to work in the region within the latitude of ±65° (Shen et al. 2018). In recent years, a few studies were published concerning the performance of different payloads of CSES. Ambrosi et al. (2018) investigated the seismo-associated perturbations of the Van Allen belts using the High Energetic Particle Detector (HEPD) of the CSES satellite mission. Concerning the performance of the Electric Field Detector (EFD) on board, Huang et al. (2018)

studied several natural electromagnetic emissions during the six-month orbit test phase, and the preliminary analysis suggested that the EFD show good performance. Cao et al. (2018) studied the data from the search coil magnetometer (SCM) mounted on CSES that was designed to measure the magnetic field fluctuation of low frequency electromagnetic waves ranging from 10 Hz to 20 kHz, they concluded that the performance of SCM can satisfy the requirement of scientific objectives of CSES mission. As one of the main payloads, the GNSS occultation receiver (GOR) had the occultation observation function of both GPS and BDS-2 (Lin et al. 2018). Yan et al. (2020) provided a comprehensive comparison of in situ electron density (Ne) and temperature (Te) measured by Langmuir probe (LAP) on board the CSES with other space-borne and ground-based observations. Their results suggested that the CSES in situ plasma parameters are reliable with a high scientific potential for the investigation of geophysics and space. Wang et al. (2019) compared CSES ionospheric RO data with Constellation Observing System for Meteorology, Ionosphere and Climate (COSMIC) measurements. Results indicated that NmF2 and hmF2 between CSES and COSMIC is in extremely good agreement, and co-located electron density profiles (EDPs) between the two sets are generally in a good agreement above 200 km.

Though the performance of CSES has already been analyzed for different payloads, there are still rooms for an in-depth analysis of GOR, especially for the region with an altitude below 200 km, e.g., E-layer. In addition, as demonstrated by previous studies, the RO measurements can provide very valuable data for the global sounding of sporadic E layers. In this study we assessed the GRO performance of CSES in the investigating of lower ionosphere, especially the occurrence and properties of sporadic E layers on a global scale.

This paper is organized as follows. We first realize the algorithm of sounding sporadic E layers with almost nine months CSES GOR data. Then, we show the results and discussions on global Es-event morphology. Afterward, the comparison of Es altitudes from RO profiles with those from Wuhan ZLT ionosonde measurements reveal a large correspondence between both measurement techniques is introduced. Finally, we present the conclusion.

## 2. Methods

The GOR payload on board CSES can receive the dual frequencies from GPS (L1: 1575.42±10 MHz; L2: 1227.6±10 MHz) and BDS-2 (L1: 1561.98±2 MHz; L2: 1207.14±2 MHz) (Wang et al, 2019). Based on GNSS RINEX format data, we calculate the electron density profile by occultation inversion algorithm (Lei et al, 2007; Yue et al, 2011), and extract the signal to noise density ratio (SNR) data of L1 and the corresponding time information according to the observation data. Considering the resolution of time and altitude, a moving average of 31 points (corresponding to 70-120 km in the vertical direction) is used to calculate the background trend term of SNR data. After that, we calculate the normalized SNR data and the standard deviation of normalized SNR data. A new criterion is developed to determine whether Es occurs. That is, when the mean value of the absolute value of the difference between the normalized SNR is greater than 3 times the standard

deviation, we consider the Es occurs. If more than one value of the normalized SNR sequence meets the conditions, multi-layer Es occurs.

## 2.1 Sounding of sporadic E layers

Signal to noise ratio, denoted as SNR or S/N (dB), which can be estimated to obtain the carrier-to-noise ratio (C/N0) measurement, provides highly desirable information about the quality of the received GNSS signal. (Gómez-Casco, D et al. 2018). The SNR is very sensitive to the electron density changing with altitude, e.g., the sporadic E layer. These vertically small variations in the electron density would lead to phase fluctuation of the GNSS signal which can be observed as a reduction or increase of the signal power at the receiver (Hajj et al., 2002). According to RINEX Version 2.10

documentation, the numerical magnitude of SNR on L1 and L2 is stored in the S1 and S2 observations in the Level-1 original observations data product of CSES, respectively.

Because SNR data itself also has a certain long-term variation, we need to extract the background trend item in SNR data to obtain the disturbance information after removing the background trend. In this study, the moving average method is used to

extract the background trend term of SNR data. The formula is as follows:

$$\overline{X_k} = \frac{X_{k-\frac{N-1}{2}} + \cdots + X_k + \cdots + X_{k+\frac{N-1}{2}}}{N} \tag{1}$$

where, $X_k$ and $\overline{X_k}$ is the k-th data of the original SNR sequence and after smoothing, and N is the size of the smoothing window. Considering that the original data processed in this study is the original occultation observation data with a sampling rate of 1Hz, so we choose 31 data points as the size of the smooth window.


It is inconvenient to analyze SNR data due to the large value of SNR data, therefore, it has to be first normalized. The calculation formula is as follows:

$$SNR1 = \frac{SNR}{SNR0} \tag{2}$$

where SNR is the original data sequence, SNR0 is the background trend item sequence, and SNR1 is the normalized data

sequence.

Note that there is no strict standard to judge whether single-layer Es or multi-layer Es occurs. In this study, 70-120 km is selected as the interval to sound the occurrence of Es events. The standard deviation of normalized SNR sequence is calculated:

$$\overline{SNR1} = \sum_{i=1}^{n} SNR1_i \tag{3}$$

$$std = \sqrt{\sum_{i=1}^{n}(SNR1_i - \overline{SNR1})^2/(n-1)} \tag{4}$$

where $\overline{SNR1}$ is the normalized SNR sequence mean; $SNR1_i$ is the normalized SNR sequence; $n$ is the number of normalized SNR sequences. It is thought that Es occurred once the difference of $SNR1_i$ from the mean is greater than 3 times the standard deviation. While there are multi-layer Es occurs in a single occultation event if multiple $SNR1_i$ meets the judgment criterion.

We selected two representative occultation events from CSES observation data as examples to verify the correctness of our Es detection algorithm. The detection of a single-layer Es event is shown in Figure 1. The left figure shows the electron density profile of G06 satellite at 06:56UT on August 14, 2018 and the SNR profile. The right figure shows the electron density profile, normalized SNR profile within 60-160 km at the same time, in which the red dotted line is $\overline{SNR1} \pm 3std$ boundary vertical line, it can be seen that there is a $SNR1_i$ whose value exceeds the boundary line and corresponds to the height of abnormal electron density in the figure. According to the normalized SNR sequence, the Es height detected in the figure is 96.49 km. The detection of multi-layer Es events is shown in Figure 2. The left figure shows the electron density profile and the SNR profile of G17 satellite at 20:58 on August 27, 2018. The right figure shows the electron density profile, normalized SNR profile within 60-160 km at the same time. The red dotted line is $\overline{SNR1} \pm 3std$ boundary vertical line, and the Es heights detected in the figure are 73.63 km and 102.76 km respectively.

Under the assumptions of spherical symmetry (i.e., assuming only vertical electron density gradients), straight-line propagation and an earth's spherical shape, we calculate the electron density profile by occultation inversion algorithm mainly referring to Lei et al. (2007). These assumptions especially the assumptions of spherical symmetry are frequently not fully accurate for smaller-scale ionospheric phenomena, the calculated electron density values are not accurate and can only describe the approximate numerical distribution. Nevertheless, this study does not attempt to retrieve the absolute accurate electron density values of Es, but shows the electron density differences at Es peaks compared to those electron density profiles without the Es phenomenon. Our new criterion is developed to mainly use the normalized SNR to determine the Es events, the electron density profile is only a reference to illustrate the effect of relatively higher electron density at Es on the normalized SNR variation, it is further verified that variance in SNR can be suggested to identify and sound sporadic E layers. There is a certain deviation in the low altitude range by these assumptions, and the electron density calculated by inversion will also have an impact. Compared with the electron density itself, the signal-to-noise ratio is more sensitive to the electron density gradient, the SNR peak height does not fully correspond to the local peak of electron density. Therefore, it will affect the inversion height comparison.

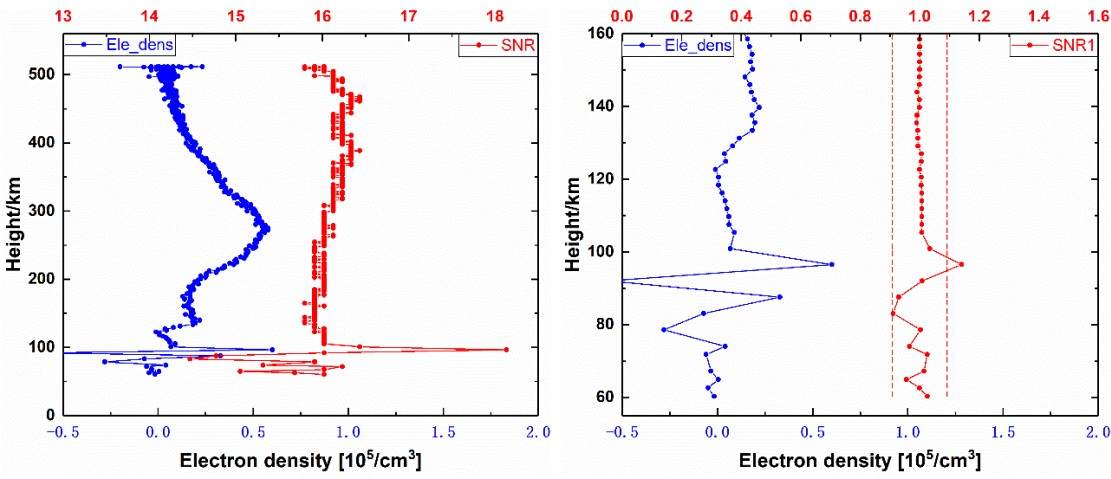

**Figure 1 Schematic diagram of G06 single-layer Es sounding. The left figure shows the electron density profile of G06 occultation event and the SNR profile at 06:56 on August 14, 2018, the right figure shows the electron density profile and normalized SNR profile within 60-160 km at the same time, and the red dotted line is $\overline{SNR1} \pm 3std$ boundary vertical.**

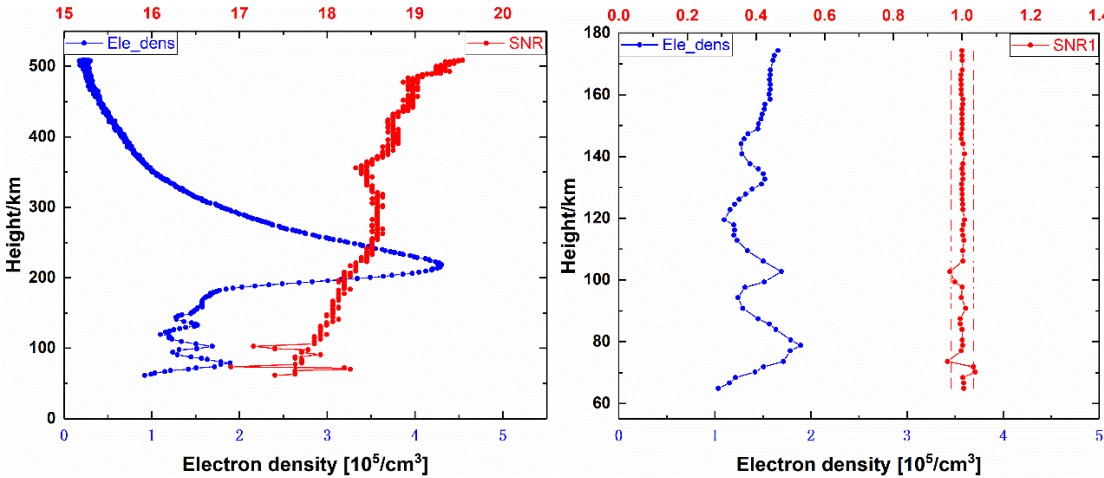

**Figure 2 Schematic diagram of G17 multi-layer Es sounding. The left figure shows the electron density profile of G17 occultation event and the SNR profile at 06:56 on August 14, 2018, the right figure shows the electron density profile and normalized SNR profile within 60-160 km at the same time, and the red dotted line is $\overline{SNR1} \pm 3std$ boundary vertical.**

## 3. Discussions on global Es-event morphology

The GOR measurement of CSES from March 1 to December 1 in 2018 are used in the data analysis. With nearly nine months of data from CSES, there are 104531 and 12642 electron density profiles obtained from GPS and BDS-2 data of CSES, respectively. The inversion algorithm is utilized based on the FUSING (FUSing IN Gnss) software (Shi et al. 2018; Zhao et al. 2018; Gu et al. 2020; Gu et al. 2021). Originally, the FUSING software is developed for high precision real-time

GNSS data processing and multi-sensor navigation, and now it can also be used for atmospheric modeling (Lou et al. 2019; Luo et al. 2020; Luo et al. 2021).

According to the orbital characteristics of CSES, the payloads of CSES mainly works in the region from 65°S to 65°N in latitude. Such as the Langmuir probe (LAP), detects the electron density in the space around the CSES satellite. As for GNSS occultation receiver (GOR), works in the region within the latitude of ±65°, but according to the principle of occultation inversion by the occultation receiver, the ionosphere that the GPS/BDS-2 satellite signals received by GOR passes through is globally distributed, the tangent points of electron density profiles from CSES are globally distributed, some scholars have given relevant global distribution results in their studies. Wang et al. (2019) showed the global distribution of the location of the tangent point of the maximum values in a profile of CSES from 90°S to 90°N. Lin et al. (2018) showed the distribution of the true NmF2, hmF2 and retrieved NmF2, hmF2 with respect to the local time and magnetic latitude from 90°S to 90°N, respectively. Cheng et al. (2018) studied that the global coverage of CSES GRO events in more than two months and compared with COSMIC observations, they concluded that both the CSES and COSMIC occultation can realize global coverage, they also showed the global distributions of layer F2 peak density and peak height derived from GRO from 90°S to 90°N.

Therefore, when we extract the electron density profiles corresponding to the tangent point and the SNR profiles data, Es occurrence rate sounded from CSES is globally distributed.

**3.1 Distribution of Es occurrence rate for seasons and altitude**

The nine-month data have been divided into spring (March, April and May), summer (June, July and August) and autumn (September, October and November). For each season, we use the altitude resolution of 1 km to count the number of occultation events which sound Es events in each altitude interval. Due to the resolution of observation values, we do not distinguish the occultation events of sounding Es in different layers. Considering the error caused by the integrity of the original observation data in different seasons and different days, we count the total number of days with observation data in each season, then calculate the ratio of the number of occultation events with Es events in different height intervals to the total number of days in the season, that is counting the number of occultation events with Es events per day. Since CSES has both GPS and BDS-2 observations, we count the average number of daily occultation events which sound Es events of different satellite systems. The results are as follows:

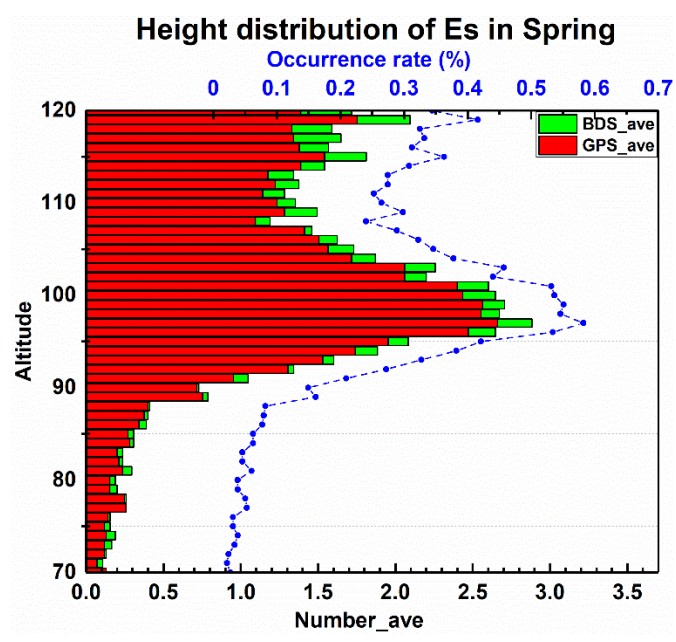

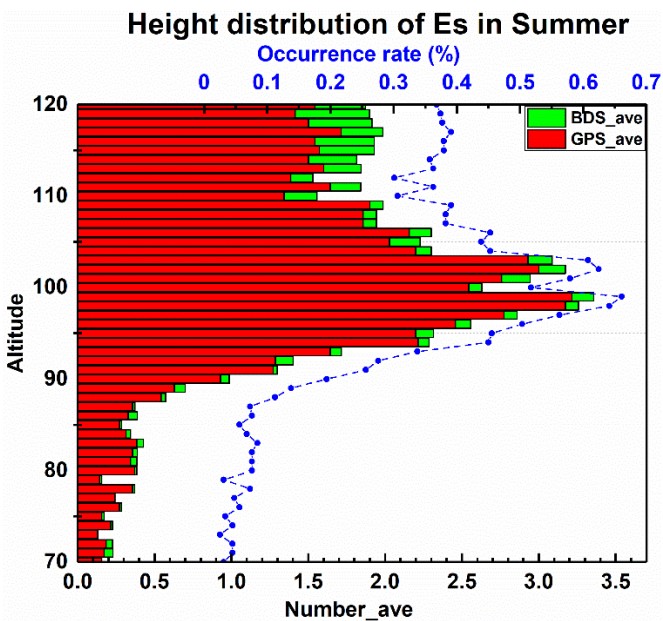

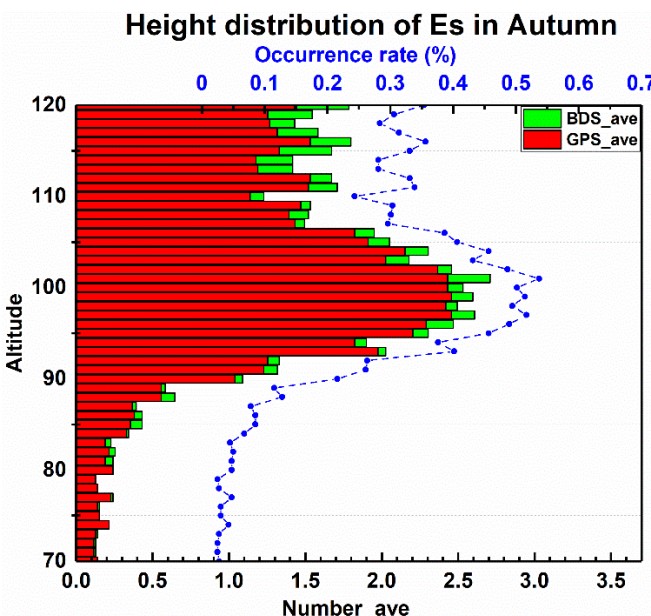

**Figure 3 Height distribution of Es average daily occurrence rate for three different seasons, from top to bottom are the results of spring, summer and autumn, respectively. The blue dotted line diagram shows Es occurrence rate, the red and green bar chart shows the number of occultation events with Es events per day.**

There are the results of spring, summer and autumn from top to bottom, respectively. Due to the lack of observation data of CSES for about 20 days in summer, it is not very appropriate to compare seasonal differences only by plotting the total number of occultation events with Es. So, as shown in the blue dotted line diagram of Fig.3, we also calculate the ratio of the number of occultation events with Es events in different height intervals to the total number of occultation events in the season. It can be seen from Fig. 3 that the Es average daily occurrence rate has obvious seasonal variation, the height of Es occurrence in spring, summer and autumn is mainly 90-110 km, the height with the largest daily average incidence of Es in spring is 98 km, with a daily average of 2.88, the height with the largest daily average incidence in summer is 99 km, with a daily average of 3.36, and in autumn the height is 101 km, with a daily average of 2.71. The results show significantly more Es appears above 110 km than below 90km overall distribution of three seasons. The reasons, firstly, there are less observation data of CSES at a lower altitude, and this situation is reflected in the blue dotted line diagram of Fig. 3; secondly, due to the time resolution, some initial lower altitude values are discarded when using the sliding window to calculate the SNR background trend term, Es occurring at a lower height is also discarded at the same time.

**3.2 Distribution of global Es occurrence rate for seasons**

The global longitude and latitude regions are divided into grids with a resolution 10°×5°. The number of occultation events in each grid and the number of occultation events with Es events are counted, and the ratio of the number of occultation

events with Es to the total number of occultation observations is taken as the Es occurrence frequency of the grid. In order to reduce the impact of accidental errors, we further optimized the statistical method, the Es occurrence rate for the grid is calculated only when the number of occultation events in the grid is greater than 10. Finally, the global longitude-latitude distribution characteristics of Es occurrence frequency in this season are obtained. The statistical results are as follows:

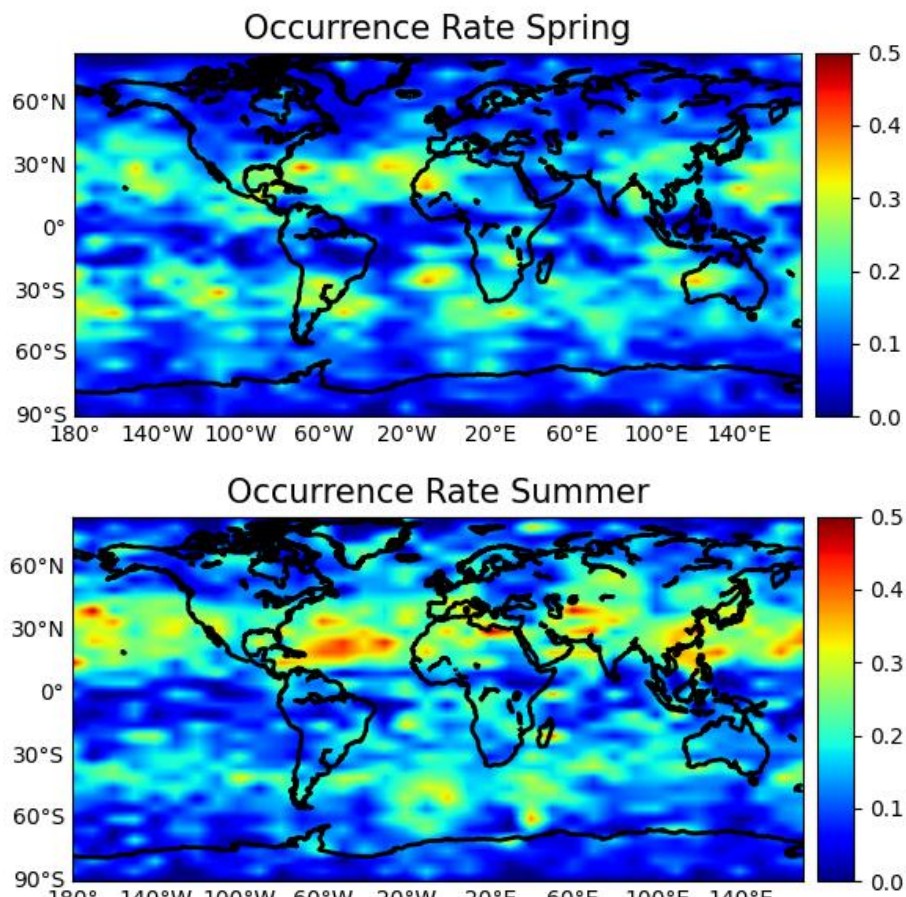

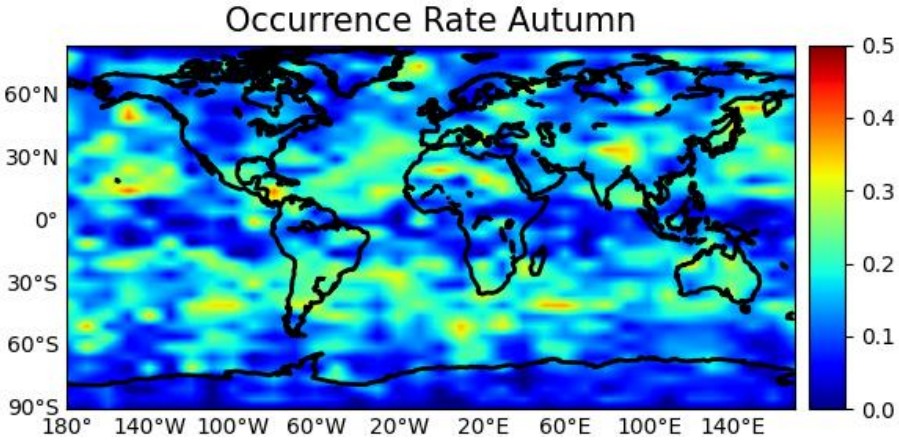


**Figure 4 The geographical distribution of Es occurrence rate for three different seasons in 5°×10° geographic latitude/longitude grid, from top to bottom are the results of spring, summer and autumn, respectively.**

There are the results of spring, summer and autumn from top to bottom, respectively. In general, Es preferably occurs at
midlatitudes of the summer hemisphere. The overall occurrence frequency of global Es in spring and autumn is lower than that in summer. This phenomenon may be due to the strong solar radiation in summer and the ionization of more metal atoms in the ionosphere, which increases the source of Es and promotes the formation of Es. Therefore, the occurrence rate in mid-latitude of the hemisphere in summer is higher than that in other latitudes (Chu et al. 2014). There is no significant difference in the frequency of Es between the northern and southern hemispheres in spring and autumn, and it shows an
almost symmetrical trend along the equator. In spring and autumn, the direct point of the sun is near the equator. Because the magnetic line of force here is almost horizontal, it is difficult to form ion aggregation even if the ionization rate increases, so the occurrence rate is relatively high in the low latitude area of the magnetic equator (Arras et al. 2017; Xue et al. 2018). The Es rates at polar regions are always low. We can also find an occurrence depression around the American area (the longitude sector of 70°–120° W) in the mid-latitude in summer, where the Es occurrence rates were lower than anywhere else along
the zone bands, this is consistent with the phenomenon found by Tsai et al. 2018.

**3.3 Distribution of Es occurrence rate for latitude and altitude**

To comprehensively analyze the distribution of Es incidence with latitude and altitude, the latitude-altitude region is divided into grids with a resolution of 10°×1km. Similarly, the ratio of the number of occultation events corresponding to Es events in the grid to the total number of days with observed data in the season is calculated, the daily average number of Es events
is taken as the occurrence frequency of Es for statistical analysis. The results are as follows:

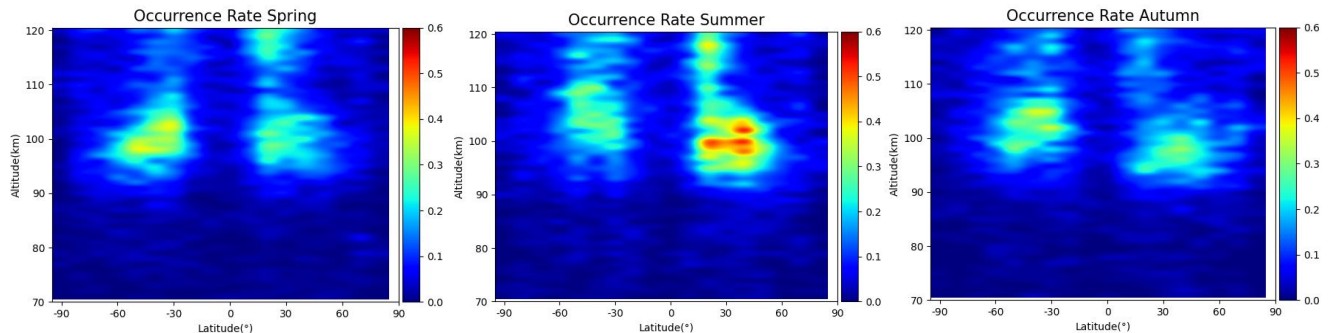

**Figure 5 The distribution of Es occurrence rate for three different seasons in 10°×1km geographic latitude/altitude grid, from left to right are the results of spring, summer and autumn, respectively.**


There are the results of spring, summer and autumn from left to right, respectively. It can be seen from the figure that the incidence of Es latitude altitude shows obvious seasonal changes. The incidence of Es in summer in the northern hemisphere is significantly higher than that in spring and autumn in the same latitude range and height range. The latitude range of Es high incidence is 20°-50° north-south latitude, mainly around 30°. The occurrence height of Es is mainly concentrated in 90-

110 km.

### 3.4 Distribution of Es occurrence rate for local time and latitude

In order to comprehensively analyze the distribution of Es incidence with local time and latitude, the local-time-latitude region is divided into grids with a resolution of 1h×5°. In order to exclude the effect of single-day observation integrity on the distribution of Es incidence with local time, we use the ratio of the number of occultation events with Es to the total

number of occultation observations in the grid, at the same time, the Es occurrence rate for the grid is calculated only when the number of occultation events in the grid is greater than 10 to reduce the impact of accidental errors. The results are as follows:

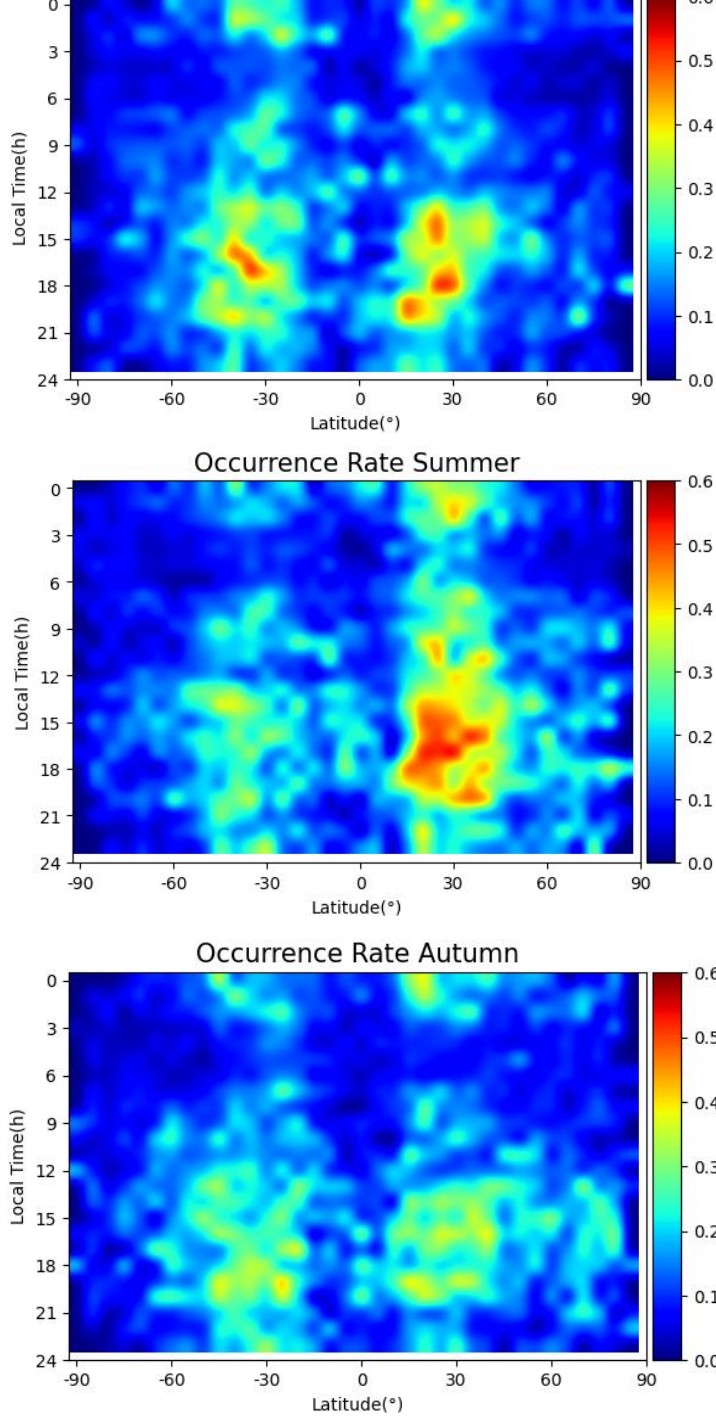


**Figure 6 The distribution of Es occurrence rate for three different seasons in 1h×5° local time/geographic latitude grid, from top to bottom are the results of spring, summer and autumn, respectively.**

There are the results of spring, summer and autumn from top to bottom, respectively. Maximum Es occurrence is expected when the zonal wind shear, which is mainly produced by the semidiurnal tide in midlatitudes (Arras et al. 2009). At midlatitudes, the Es activity is dominated primarily by a semidiurnal feature, which is generally believed to be induced by east–west zonal winds in terms of semidiurnal tides, especially in spring and summer (Whitehead 1989; Chu et al. 2014). The semidiurnal tides generally start around 6 and 14 LT, continue for 14 h, and then fade out around 20 and 4 LT separately (Tsai et al. 2018). So, it can be seen from the figure that the incidence of Es shows obvious local time changes, the period of local time 14:00-20:00 is the high incidence period of Es.

## 4. Experiments of Comparing with Ionosonde measurements

In this study, we choose a certain space-time matching criterion to obtain the pairs of the geometric altitudes of a sporadic E layer detected in CSES radio occultation profiles and the virtual heights of a sporadic E layer obtained by the ZLT ionosonde for confirming the derived sporadic E parameter in height. Luo et al. (2019) choose a certain space-time matching criterion to evaluate the quality of the electron density profile from the FY-3C mission with respect to COSMIC mission. We modified their method to confirm the height of the derived sporadic E layer. We counted the data of Wuhan ZLT ionosonde from March 1 to December 16 in 2018 of the same period, and extracted the h'Es data. The space-time matching criterion is quantified as the size of the space-time window centered on the position and occurrence time of the sporadic E layer obtained by the ZLT ionosonde. The sporadic E layer detected in CSES radio occultation profiles falling into the space-time window and the sporadic E layer obtained by the ZLT ionosonde constitute the pairs participating in the comparative analysis. Here the space-time window is denoted as (B, L, T), where B and L represent the size of space window along latitude and longitude, respectively; T represents the size of the time window.

In this study, considering that the temporal resolution of the ionosonde is 15 minutes, four different space-time matching criteria are proposed with the window as (10°, 10°, 7.5 min), (5°, 10°, 7.5min), and (5°, 5°, 7.5min), respectively. Among the other parameters, the height of sporadic E layer is an important parameter of the derived sporadic E layer. Thus, the correlation coefficient (CC), is derived for determining the height of sporadic E layer. The definition of the correlation coefficient is presented below.

$$CC = \frac{\sum_{i=1}^{N}\left(x_i^C \cdot x_i^Z\right) - \frac{1}{N}\sum_{i=1}^{N} x_i^C \sum_{i=1}^{N} x_i^Z}{\sqrt{\left(\sum_{i=1}^{N}(x_i^C)^2 - \frac{1}{N}(\sum_{i=1}^{N} x_i^C)^2\right)\left(\sum_{i=1}^{N}(x_i^Z)^2 - \frac{1}{N}(\sum_{i=1}^{N} x_i^Z)^2\right)}} \tag{6}$$

where N represents the total number of data pairs in the matching group under a given spatiotemporal matching windows; $X_i^C$ ($i=1,2,3\cdots,n$) represents the geometric altitudes of $i$th sporadic E layer detected in CSES radio occultation profiles; $X_i^Z$ ($i=1,2,3\cdots,n$) represents the virtual heights of $i$th sporadic E layer obtained by the ZLT ionosonde.

The data of ionosonde is stored in SAO format files, this data file contains different types of parameters, such as station information and detection time, ionospheric characteristic parameters for automatic measurements, echo traces (virtual height, amplitude, doppler, frequency) at different height layers of the ionosphere (E, F1, F2), electron density profiles, virtual height and critical frequency of Es trace, etc. The SAO format description can refer to https://ulcar.uml.edu/~iag/SAO-4.htm. In order to facilitate the reading and use of data, SAOExplorer software (http://ulcar.uml.edu/SAO-X/SAO-X.html) has been developed by the Center for Atmospheric Research at the University of Massachusetts Lowell, USA, to display and measure Digisonde Ionospheric frequency map observed by a series of ionospheric altimeters.

Figure 7 shows an example of simultaneous detecting Es by CSES and ZLT ionosonde,the top left figure shows the electron density profile and the SNR1 profile of G27 satellite at 17:42 on May 17, 2018, and the electron density profile of ZLT ionosonde at 17:45 on May 17, 2018, the top right figure shows the electron density profile of ZLT ionosonde and the SNR1 profile in the range of 0-220 km. In the figure, the geodetic coordinates of Es detected by CSES is (33.0°N, 112.3°E, 99.2 km) and the geodetic coordinates of Es detected by ZLT is (30.5°N, 114.4°E, 102.5 km). The bottom figure shows the ionogram image of Wuhan ZLT ionosonde to show Es situation. We can obtain the virtual height of Es is 102.5km, we can also obtain Es layer critical frequency and frequency map at about 100km.

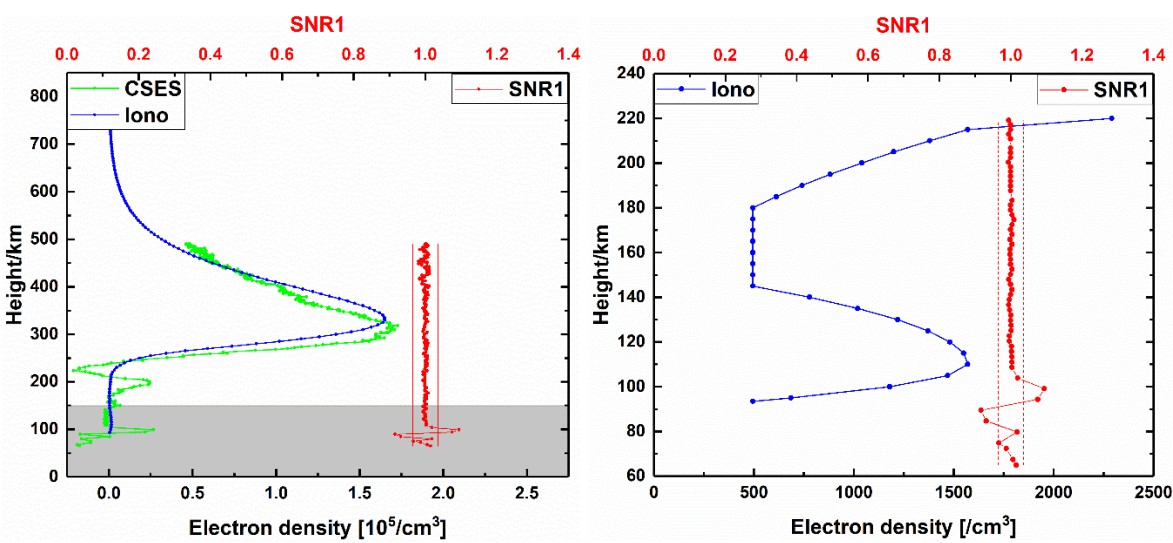

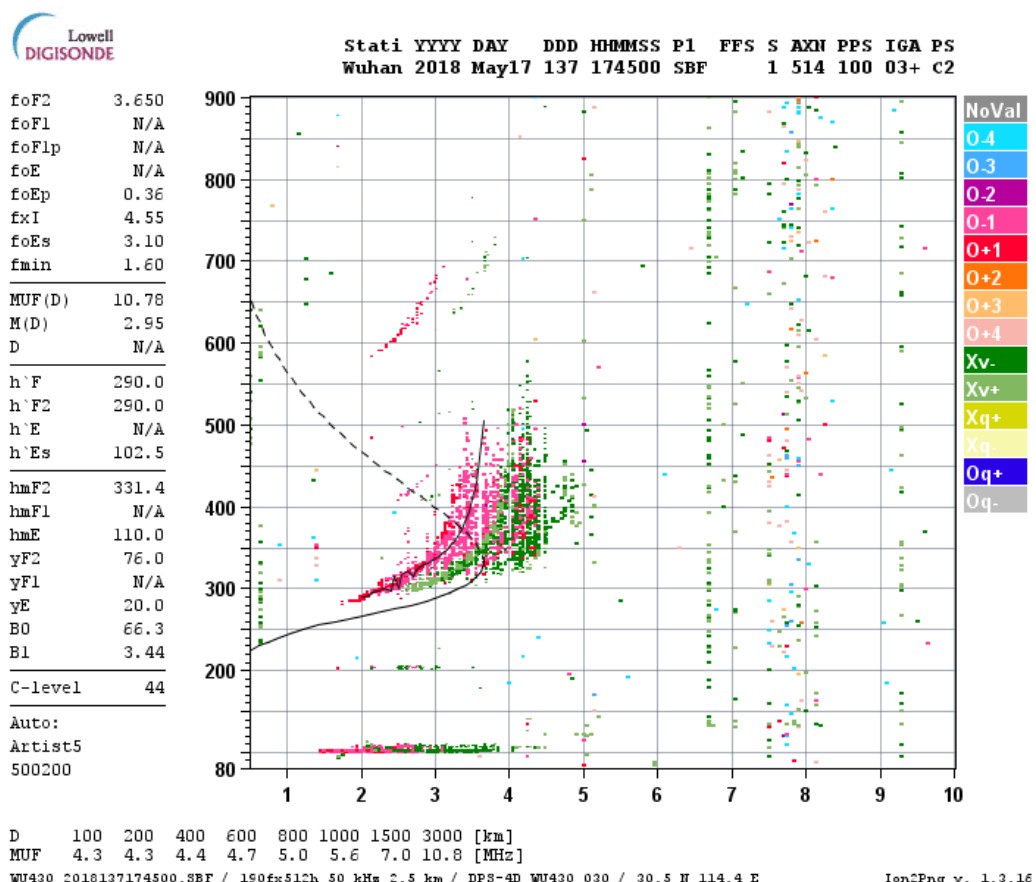

Figure 7 An example of simultaneous detecting Es by CSES and ZLT ionosonde, the top left figure shows the electron density profile and the SNR1 profile of G27 satellite at 17:42 on May 17, 2018, and the electron density profile of ZLT ionosonde at 17:45 on May 17, 2018, the top right figure shows the electron density profile of ZLT ionosonde and the SNR1 profile in the range of 0-220 km. The bottom figure shows the ionogram image of Wuhan ZLT ionosonde.

Figure 8 presents the comparison of the geometric altitudes of a sporadic E layer detected in CSES radio occultation profiles and the virtual heights of a sporadic E layer obtained by the ZLT ionosonde. We also show the regression line as the solid black line and corresponding statistical coefficients in every subgraph. These figures reveal that there is a good agreement between both parameters, which can also be seen from the high correlation larger than 0.7. The comparison among different windows conclude that the correlation increased slightly as a stricter space-time matching window involved, but with less pairs or couples. Compared with results from Arras et al. (2017), we also found a height offset between both measurement techniques mainly concentrated in 100-110 km of ionosonde altitude, and the calculation results of different space-time windows are different. The mean offset values in 100-110km are 2.36km, 2.25km, and 2.90km, which correspond to space-time matching windows (10°, 10°, 7.5 min), (5°, 10°, 7.5min), and (5°, 5°, 7.5min), respectively. This may result from the different height parameters used for both techniques: the geometric heights provided by the RO technique and the virtual

height which is influenced by the ionization below the sporadic E layer calculated from ionosonde recordings (Arras et al. 2017).

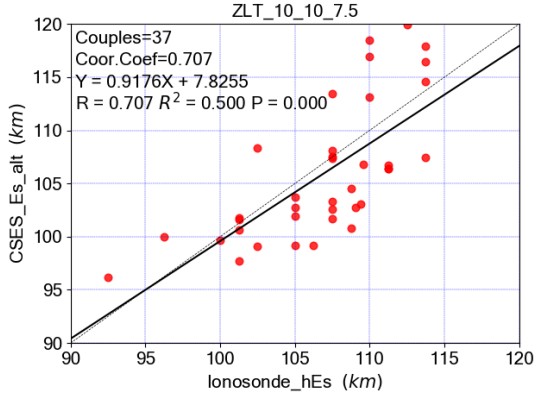

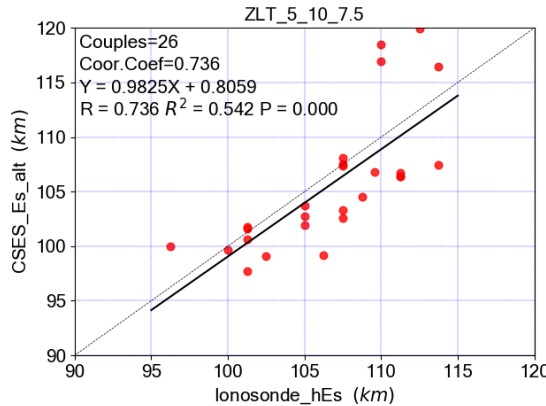

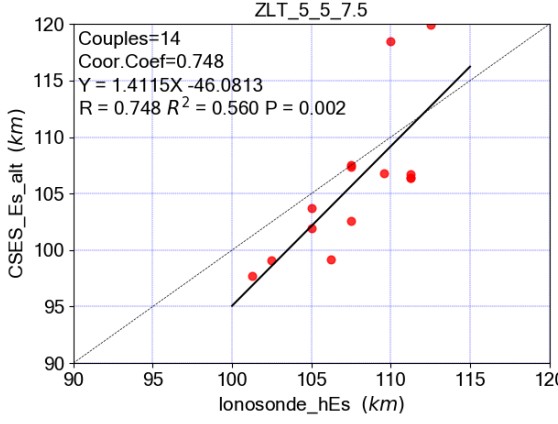

**Figure 8 Comparison of the geometric altitudes of Es detected in CSES radio occultation profiles and the virtual heights of Es obtained by the ZLT ionosonde. From top to bottom are the results of space-time matching window (10°, 10°, 7.5 min), (5°, 10°, 7.5min) and (5°, 5°, 7.5min), respectively. The black solid line is the regression line.**

## 5. Conclusions

The RO plays an important role in sounding of sporadic E layers. As China's first electromagnetic satellite, CSES has already provided service for more than three years up to now. In this study, the Level-1 data of CSES and Wuhan ZLT ionosonde from March 1 to December 1 in 2018 are collected in sounding of sporadic E layers used to study the comparison of heights.

We calculate the geodetic longitude; latitude and elevation of each occultation tangent point in the occultation inversion process, and count the corresponding time information, then extract the SNR data of L1 observations in the occultation inversion period. The occurrence of Es is judged according to the judgment criterion of $|SNR1_i - \overline{SNR1}| > 3std$. Single layer or multi-layer Es is judged according to the number of data whose sequence meets the judgment criterion. Combined with the electron density profile of occultation inversion, the correctness of our Es detection algorithm is verified.

According to the Es results we detected, we drew distribution of Es occurrence rate for seasons and altitude, distribution of global Es occurrence rate for seasons. It is concluded that the occurrence height of Es is mainly located at 90-110 km, and there are obvious seasonal and latitudinal changes in the occurrence rate of Es. There is no significant difference in the occurrence frequency of Es in the northern and southern hemispheres in spring and autumn, and it is almost symmetrical

along the equator. Summer in the northern hemisphere is the time period of high incidence of Es, and the latitude range of high incidence of Es is 20°-50° in the northern and southern latitudes, mainly around 30°. The period of local time 14:00-20:00 is the high incidence period of Es.

Finally, the comparison of the geometric altitudes of sporadic E layers detected in CSES radio occultation profiles and the

virtual heights of sporadic E layers obtained by the ZLT ionosonde was carried out with for different space-time matching window, i.e., (10°, 10°, 7.5 min), (5°, 10°, 7.5min), and (5°, 5°, 7.5min). For these four windows, the number of CSES matched pairs was 37, 26, and 14, respectively. The correlation coefficients of altitudes were 0.707, 0.736, and 0.748, respectively. The comparison of Es altitudes from RO profiles with those from coinciding ground based ionosonde measurements revealed a large correspondence between both measurement techniques.

## Data availability

CSES Radio Occultation data can be downloaded from http://www.leos.ac.cn and email author Shengfeng Gu. The Wuhan ZLT ionosonde observations can be downloaded from https://data.meridianproject.ac.cn/.

## Author contributions

XL, CX and SG designed the research; CG and JH performed the research; CG, JH and SG analyzed the data; CG drafted the paper. XL, CX and SG put forward valuable modification suggestions. All authors contributed by providing the necessary data and discussions and writing the paper.

## Competing interests

The authors declare that they have no conflict of interest.

## Acknowledgements

CSES Radio Occultation data can be downloaded from http://www.leos.ac.cn. The authors express their thanks. We also acknowledge the use of data of Wuhan ZLT ionosonde from the Chinese Meridian Project.

## Financial support

This research has been supported by the National Key R&D Program of China (grant no. 2018YFC1503502). This work is also supported by the National Natural Science Foundation of China (No. 42104029).

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
