# Peer review of "Sounding of sporadic E layers from CSES radio occultation and comparing with ionosonde measurements"

_Annales Geophysicae, 2022_

## Author Comment (AC1)

RC1:
General comments:

The paper deals with radio occultation measurement of sporadic layers. It presents Es occurrence rate depending on seasons (Spring, Summer, Autumn) and heights (70 km to 120 km), and global distribution of Es during different seasons. Included is a chapter which compares radio occultation with ionosonde measurement. The paper is well and clearly written. I have following suggestions and questions to the authors:

Thank you very much for your positive suggestions and careful reminding on our manuscript to improve the quality of it.

1. Figure 3 and corresponding text deals with histograms of Es occurrence depending on height. The plots show number of Es observations in each height bin (resolution 1 km). My question is: is it possible to present the data as percentage of Es observation / total of measurement rather than absolute numbers? The authors discuss decreased number of Es observations in summer due to lack of data. I think that using relative number rather than absolute number can help with this issue.

Thanks for your nice suggestions, your suggestion really means a lot to us. Yes, using relative number rather than absolute number can help with this issue. We will change this in the revised manuscript. Thank you again for your point out.

2. Figure 7 shows electron density profiles by CSES and ZLT ionosonde. Could you please explain if the Es can be seen in the ionosonde derived profile (I cannot tell if the peak is Es or E layer) and if yes, give details about the electron density computation? Regarding this, I strongly suggest that you show ionogram which shows Es situation, not only computed electron density profiles.

Thanks for your reminding. We are sorry for the misunderstanding caused by Figure 7. In order to better tell the peak is Es rather than E layer, we reselect an example to illustrate the situation of Es. The data of ionosonde is stored in SAO format files, the virtual height of Es and electron density profiles can be obtained directly in the SAO format files of ZLT ionosonde.
Thanks for your nice suggestions. we will add the ionogram image of Wuhan ZLT ionosonde to show Es situation more clearly.
We will include them in the revised manuscript.

3. The authors claim that the virtual height of Es can be influenced by the ionization of the ionosphere below Es. Can you estimate by how much can the h'Es theoretically differ from real height of Es for your situation?

Thanks for your suggestions. The height offset is mainly concentrated in 100-110km of ionosonde altitude, and the calculation results of different space-time windows are

different. The mean offset values in 100-110km are 2.36km, 2.25km, 2.90km, and 3.09km, which correspond to space-time matching windows (10°, 10°, 7.5 min), (5°, 10°, 7.5min), (5°, 5°, 7.5min) and (2°, 5°, 7.5min), respectively. We will include them in the revised manuscript.

4. Could you please provide brief information about the ionosonde used and software which computes electron density profile?

Thanks for your suggestions. The data of ionosonde is stored in SAO format files, this data file contains different types of parameters, such as station information and detection time, ionospheric characteristic parameters for automatic measurements, echo traces (virtual height, amplitude, doppler, frequency) at different height layers of the ionosphere (E, F1, F2), electron density profiles, virtual height and critical frequency of Es trace, etc. The SAO format description can refer to https://ulcar.uml.edu/~iag/SAO-4.htm. So, we directly read the electron density profile and plotted the image.

As for related software, in order to facilitate the reading and use of data, SAOExplorer software (http://ulcar.uml.edu/SAO-X/SAO-X.html) has been developed by the Center for Atmospheric Research at the University of Massachusetts Lowell, USA, to display and measure Digisonde Ionospheric frequency map observed by a series of ionospheric altimeters.

We will include them in the revised manuscript.

5. Figure 8 shows comparison of radio occultation Es heights vs. ionosonde derived heights. It shows a line y=x. In first two panels I had an impression that it is a regression line. I suggest that you show the regression line and corresponding statistical coefficients describing the regression line.

Thanks for your valuable suggestions. Our original purpose of drawing the line y=x is to facilitate the reader to compare the degree of deviation of radio occultation Es heights and ionosonde derived heights from y=x. Yes, it would be more understandable if we show the regression line and corresponding statistical coefficients describing the regression line. Thank you again for your suggestions to improve the quality of our manuscript. We will include them in the revised manuscript.

Small changes:

Please change "we first calculates...", and "then extracts..." in page 3.

Thanks for your careful checks. We will change them in the revised manuscript.

---

## Author Response (AR1)

**Response to Reviewers**

Dear Editor and Reviewers:

Thank you very much for your detailed suggestions, which greatly help us to improve the content and quality of manuscript. We have taken into consideration the comments by the referees and revised the manuscript carefully. Our revisions within the manuscript are highlighted in yellow color to assist the reviewers. We hope we have solved all the comments. The point-by-point replies and a list of changes are included in this document.

**RC1:**

General comments:

The paper deals with radio occultation measurement of sporadic layers. It presents Es occurrence rate depending on seasons (Spring, Summer, Autumn) and heights (70 km to 120 km), and global distribution of Es during different seasons. Included is a chapter which compares radio occultation with ionosonde measurement. The paper is well and clearly written. I have following suggestions and questions to the authors:

1. Figure 3 and corresponding text deals with histograms of Es occurrence depending on height. The plots show number of Es observations in each height bin (resolution 1 km). My question is: is it possible to present the data as percentage of Es observation / total of measurement rather than absolute numbers? The authors discuss decreased number of Es observations in summer due to lack of data. I think that using relative number rather than absolute number can help with this issue.

**Our reply:** Thanks for your nice suggestions, your suggestion really means a lot to us. Yes, using relative number rather than absolute number can help with this issue. We change this in lines 206-210 and Figure 3 in the marked-up version of revised manuscript. Thank you again for your point out.

2. Figure 7 shows electron density profiles by CSES and ZLT ionosonde. Could you please explain if the Es can be seen in the ionosonde derived profile (I cannot tell if the peak is Es or E layer) and if yes, give details about the electron density computation? Regarding this, I strongly suggest that you show ionogram which shows Es situation, not only computed electron density profiles.

**Our reply:** Thanks for your reminding. We are sorry for the misunderstanding caused by Figure 7. In order to better tell the peak is Es rather than E layer, we reselect an example to illustrate the situation of Es. The data of ionosonde is stored in SAO format files, the virtual height of Es and electron density profiles can be obtained directly in the SAO format files of ZLT ionosonde.

Thanks for your nice suggestions. We add the ionogram image of Wuhan ZLT

ionosonde to show Es situation more clearly.

We include them in lines 335-354 and Figure 7 in the marked-up version of revised manuscript.

3. The authors claim that the virtual height of Es can be influenced by the ionization of the ionosphere below Es. Can you estimate by how much can the h'Es theoretically differ from real height of Es for your situation?

**Our reply:** Thanks for your suggestions. The height offset is mainly concentrated in 100-110km of ionosonde altitude, and the calculation results of different space-time windows are different. The mean offset values in 100-110km are 2.36km, 2.25km and 2.90km, which correspond to space-time matching windows (10°, 10°, 7.5 min), (5°, 10°, 7.5min), (5°, 5°, 7.5min), respectively. We include them in lines 363-366 in the marked-up version of revised manuscript.

4. Could you please provide brief information about the ionosonde used and software which computes electron density profile?

**Our reply:** Thanks for your suggestions. The data of ionosonde is stored in SAO format files, this data file contains different types of parameters, such as station information and detection time, ionospheric characteristic parameters for automatic measurements, echo traces (virtual height, amplitude, doppler, frequency) at different height layers of the ionosphere (E, F1, F2), electron density profiles, virtual height and critical frequency of Es trace, etc. The SAO format description can refer to https://ulcar.uml.edu/~iag/SAO-4.htm. So, we directly read the electron density profile and plotted the image.

As for related software, in order to facilitate the reading and use of data, SAOExplorer software (http://ulcar.uml.edu/SAO-X/SAO-X.html) has been developed by the Center for Atmospheric Research at the University of Massachusetts Lowell, USA, to display and measure Digisonde Ionospheric frequency map observed by a series of ionospheric altimeters.

We include them in lines 320-327 in the marked-up version of revised manuscript.

5. Figure 8 shows comparison of radio occultation Es heights vs. ionosonde derived heights. It shows a line y=x. In first two panels I had an impression that it is a regression line. I suggest that you show the regression line and corresponding statistical coefficients describing the regression line.

**Our reply:** Thanks for your valuable suggestions. Our original purpose of drawing the line y=x is to facilitate the reader to compare the degree of deviation of radio occultation Es heights and ionosonde derived heights from y=x. Yes, it would be more understandable if we show the regression line and corresponding statistical coefficients describing the regression line. Thank you again for your suggestions to improve the quality of our manuscript. We include them in lines 357-358 and in Figure 8 in the

marked-up version of revised manuscript.

**Small changes:**

Please change "we first calculates...", and "then extracts..." in page 3.

**Our reply:** Thanks for your careful checks. We change them in lines 92-93 in the marked-up version of revised manuscript.

**RC2:**

The paper deal with the detection of sporadic E (Es) layers on a global scale applying the radio occultation (RO) technique. For their study, the authors use data obtained from the Chinese CSES mission. The authors developed a new algorithm to detect sporadic E signatures from RO profiles. The results show that Es appears mainly at heights between 90-110km and preferably in the summer hemisphere in the local daytime hours. The comparison with co-located ionosonde measurements shows a relatively high correlation between both measurements techniques.

The results more or less confirm what we know about sporadic E layer occurrence from former global studies. The paper does not provide new knowledge on the Es phenomenon. Nevertheless, I support the publication of the manuscript after a careful revision since it introduces the valuable and widely unknown RO data set of the CSES satellite to the community. Please find my detailed comments below.

- It would be informative to add some more details about the CSES satellite. At which altitude and inclination is it flying?

    **Our reply:** Thank you for your suggestion. We feel sorry that we did not provide enough information about CSES. The CSES is a 3-axes-stabilized satellite, based on the Chinese CAST2000 platform, with a mass of about 730 kg and peak power consumption of about 900 W. Scientific data are transmitted in the X-Band at 120 Mbps. The orbit is circular Sun-synchronous, at an altitude of about 507 km, inclination of about 97.4°, descending node at 14:00 LT.
    We include this in lines 59-63 in the marked-up version of revised manuscript.

- Do the RO profiles cover the whole globe? Are the data equally distributed in local time?

    **Our reply:** Thank you for your reminding. All payloads of CSES are designed to work in the region within the latitude of ±65°, and the RO profiles range of CSES is mainly between −65° and +65° of geographic latitudes. Overall, the GRO occultation can realize global coverage, we can refer to Cheng et al, 2018 (http://doi.org/10.26464/epp2018048). The GOR measurement of CSES from March 1 to December 1 in 2018 are used in the data analysis. With nearly nine months of data from CSES, there are 104531 and 12642 electron density profiles obtained from GPS and BDS-2 data of CSES, respectively. The original data with a sampling rate of 1Hz is from nearly 0:00 to nearly 24:00 every day. When we analyze the distribution of Es incidence with local time and latitude as the section 3.4, although we did not separately provide the distribution of Es incidence with local time, the original data and RO profiles are equally distributed in local time.

- Could you add some information to the "Methods" section about the altitude resolution of your RO profiles? What is the signal tracking frequency?

**Our reply:** Thanks for your suggestions. Based on GNSS RINEX format data, we calculate the electron density profile by occultation inversion algorithm (Lei et al, 2007; Yue et al, 2011). The original data processed in this study is the original occultation observation data with a sampling rate of 1Hz. So, the time resolution of our RO profiles is 1Hz. According to occultation inversion algorithm, we can get RO electron density profiles, but we can't easily get the altitude resolution directly. As shown in Figure 1-2, the altitude of RO profiles is unequally distributed. So, we feel sorry that we could not provide precise altitude resolution of our RO profiles.

The GOR payload on board CSES can receive the dual frequencies from GPS (L1: $1575.42\pm10$ MHz; L2: $1227.6\pm10$ MHz) and BDS-2 (L1: $1561.98\pm2$ MHz; L2: $1207.14\pm2$ MHz). We include this in lines 91-92 in the marked-up version of revised manuscript, thank you.

- Figures 4-6 are my major point of criticism: Due to the relatively low amount of RO profiles, the plots 4-6 are not very informative and deviate distinctly from existing global Es plots. I recommend increasing the grid size slightly (maybe 10° in longitude) or working with sliding windows of a bigger size.

**Our reply:** Thanks for your valuable recommendation. We increase the grid size of 10° in longitude to analyze the global Es-event distribution and morphology. In order to reduce the impact of accidental errors, we further optimized the statistical method, the Es occurrence rate for the grid is calculated only when the number of occultation events in the grid is greater than 10. We change them in Figures 4-6 in the marked-up version of revised manuscript.

- line 240-243: There is a contradiction between figure 6 and the text. In the text, you write that the high incidence of Es in the local afternoon is related to high solar radiation. In Fig. 4 (summer plot) the values of Es occurrence are of the same magnitude at 3-6 in the morning where there is definitely no sunshine. Could these high early morning values simply be relicts from data availability? Is an effect from the wind possible? Please comment on it.

**Our reply:** Thanks for your valuable reminding. We are sorry for our careless mistakes. When we calculate the number of GNSS RO events of each day for CSES GPS, CSES BDS-2, we find the daily RO events number has a certain deviation. Then we checked the RO profiles and the original observations. For the full-day original observations, the RO profiles are uniformly distributed at the local time, but there are some incomplete original observations, especially close to the initial operation period of CSES launch, therefore, the distribution of RO profiles at the local time in those days is not uniform, there is a certain deviation in further obtaining the distribution of the number of Es occurrence at the local time. In the original manuscript, we used the ratio of the number of occultation events with Es events in the grid to the total number of days to calculate the occurrence rate, which caused a very large error. We use a new

calculation, that is the ratio of the number of occultation events with Es to the total number of occultation observations in the grid, at the same time, the Es occurrence rate for the grid is calculated only when the number of occultation events in the grid is greater than 10 to reduce the impact of accidental errors. After calculation, there is no such abnormal phenomenon at 3-6 in the morning.

We include these in lines 274-278 and in Figures 6 in the marked-up version of revised manuscript.

Thank you again for your positive comments and valuable suggestions to improve the quality of our manuscript.

- You show electron density profiles obtained from RO measurements. These profiles are frequently not accurate for smaller-scale ionospheric phenomena since they rely on assumptions like spherical symmetry which is not valid for sporadic E. Could you comment a bit on the assumptions used for calculating the electron density profiles here?

**Our reply:** Thank you for suggestions. Under the assumptions of spherical symmetry (i.e., assuming only vertical electron density gradients), straight-line propagation and an earth's spherical shape, we calculate the electron density profile by occultation inversion algorithm mainly referring to Lei et al. (2007) (https://doi.org/10.1029/2006JA012240).

Yes, you are right, these assumptions especially the assumptions of spherical symmetry are frequently not fully accurate for smaller-scale ionospheric phenomena, the calculated electron density values are not accurate and can only describe the approximate numerical distribution. Nevertheless, this study does not attempt to retrieve the absolute accurate electron density values of Es, but show the electron density differences at Es peaks compared to those electron density profiles without the Es phenomenon.

Our new criterion is developed to mainly use the normalized SNR to determine the Es events, the electron density profile is only a reference to illustrate the effect of relatively higher electron density at Es on the normalized SNR variation, it is further verified that variance in SNR can be suggested to identify and sound sporadic E layers.

We comment a bit on the assumptions in lines 152-161 in the marked-up version of revised manuscript.

- Figure 8: I assume the black line is no regression but x=y line, correct? It is a little bit misleading since there is definitely an offset between both parameters simply because virtual heights are always deviating from geometric ones.

**Our reply:** Thank you for your reminding. Yes, the black line is no regression but x=y line. Our original purpose of drawing the line y=x is to facilitate the reader to compare the degree of deviation of radio occultation Es heights and ionosonde derived heights from y=x. We show the regression line and corresponding statistical coefficients in Figure 8 in the marked-up version of revised manuscript.

What is the mean offset between both techniques and can this be explained by different altitude systems?

**Our reply:** According to our experimental results, the height offset is mainly concentrated in 100-110km of ionosonde altitude, and the calculation results of different space-time windows are different. The mean offset values in 100-110km are 2.36km, 2.25km, 2.90km, and 3.09km, which correspond to space-time matching windows (10°, 10°, 7.5 min), (5°, 10°, 7.5min), (5°, 5°, 7.5min) and (2°, 5°, 7.5min), respectively. We include this in lines 363-366 in the marked-up version of revised manuscript.

In the lower right plot, there should be 5 couples. I only see 4. Is it convincing enough to calculate a correlation coefficient from 4-5 values only?

**Our reply:** Thanks for your careful checks. There really be 5 couples, two of the points are very close in height and almost overlap in the figure, we have commented on line 358-360. Yes, it is really not suitable to calculate the correlation coefficient with only 5 values, our original purpose of calculating this space-time matching windows is to draw conclusions that the correlation increased slightly as a stricter space-time matching window involved, but with less pairs or couples. The first three space-time matching window figures have revealed that there is a good agreement between both parameters, which can also be seen from the high correlation larger than 0.7. After discussions with our authors, we decided to remove the results of this window in the revised manuscript, thank you again.

- Please carefully revise the complete references section. There are many typos and different styles in citing existing literature.

**Our reply:** We are really sorry for our careless mistakes. Thank you for your reminding. We carefully revise the complete references section in the revised manuscript.

**small improvements:**

line 185-186. I assume there is a detail missing in this sentence. For me, it is hard to follow your intention.

**Our reply:** Thanks for your careful checks. We are sorry for our carelessness. We rewrite them in line 216 in the marked-up version of revised manuscript.

line 201, 222, and 239: ...results with spring.... "with" is not the correct word here. Please reformulate.

**Our reply:** Thanks for your careful checks. We correct 'with' to 'of' in line 206, 240, 265, and 292 in the marked-up version of revised manuscript.

---

## Author Response (AR2)

**Response to Reviewers**

Dear Editor and Reviewers:

Thank you very much for your detailed suggestions, which greatly help us to improve the content and quality of manuscript. We have taken into consideration the comments by the referees and revised the manuscript carefully. Our revisions within the manuscript are highlighted in yellow color to assist the reviewers. We hope we have solved all the comments. The point-by-point replies and a list of changes are included in this document.

**RC1:**

The authors answered my question and comments therefore I agree with the publication of the manuscript. I suggest careful final reading. For example, in captions of Figures 5 and 6 I found " are the results with of spring, summer and autumn...". In page 11, the sentence "In order to exclude the effect of single-day observation integrity on the distribution of Es incidence with local time." should be followed by some part of text.

**Our reply:** Thanks for your careful checks. We change captions of Figures 5 and 6 in lines 249-250 and 271-272 in the marked-up version of revised manuscript. In page 11, We rewrite them in lines 262-266 in the marked-up version of revised manuscript. Thank you again for your point out.

I find following sentence somewhat shallow or misleading, especially the mentioning of "electricity": "This is because the occurrence of Es is directly related to the intensity of solar radiation, the electricity and aggregation of metal ions gradually occur, therefore, there is a delay between the high incidence period of Es and the 12:00 295of local time with the strongest solar radiation." (page 13). I suggest to include more detailed explanation of the electric field orientation leading to increase in Es formation.

**Our reply:** We are sorry for our careless mistakes and the misleading we caused. Thank you for your suggestion. We restate the results in Figure 6 and include more detailed explanation of Es formation.

Maximum Es occurrence is expected when the zonal wind shear, which is mainly produced by the semidiurnal tide in midlatitudes (Arras et al. 2009). At midlatitudes, the Es activity is dominated primarily by a semidiurnal feature, which is generally believed to be induced by east–west zonal winds in terms of semidiurnal tides, especially in spring and summer (Whitehead 1989; Chu et al. 2014). The semidiurnal tides generally start around 6 and 14 LT, continue for 14 h, and then fade out around 20 and 4 LT separately (Tsai et al. 2018). So, it can be seen from the figure that the incidence of Es shows obvious local time changes, the period of local time 14:00-20:00 is the high incidence period of Es.

We rewrite them in lines 280-286 in the marked-up version of revised manuscript. Thank you again for your suggestions to improve the quality of our manuscript.

**RC2:**

The authors made good progress in arranging their manuscript. However, there are still some questions remaining.

There is one major point of concern:
You stated in the text that the CSES satellite provides data stretching from 65°S to 65°N in latitude. But Figure 4-6 show data up to 90°N/S. Please check your data and your gridding. You said that you are using a 5° latitudinal gridding. Thus, it is simply impossible to get a data coverage up to 90°.
Please double-check your data and gridding!

**Our reply:** According to the orbital characteristics of CSES, the payloads of CSES mainly works in the region from 65°S to 65°N in latitude. Such as the Langmuir probe (LAP), detects the electron density in the space around the CSES satellite. As for GNSS occultation receiver (GOR), works in the region within the latitude of ±65°, but according to the principle of occultation inversion by the occultation receiver, the ionosphere that the GPS/BDS-2 satellite signals received by GOR passes through is globally distributed, the tangent points of electron density profiles from CSES are globally distributed, some scholars have given relevant global distribution results in their studies.

Wang et al. (2019) (https://doi.org/10.5194/angeo-37-1025-2019) showed the global distribution of the location of the tangent point of the maximum values in a profile of CSES from 90°S to 90°N. Lin et al. (2018) (https://doi.org/10.1007/s11431-018-9245-6) showed the distribution of the true NmF2, hmF2 and retrieved NmF2, hmF2 with respect to the local time and magnetic latitude from 90°S to 90°N, respectively. Cheng et al. (2018) (http://doi.org/10.26464/epp2018048) studied that the global coverage of CSES GRO events in more than two months and compared with COSMIC observations, they concluded that both the CSES and COSMIC occultation can realize global coverage, they also showed the global distributions of layer F2 peak density and peak height derived from GRO from 90°S to 90°N.

Therefore, when we extract the electron density profiles corresponding to the tangent point and the SNR profiles data, Es occurrence rate sounded from CSES is globally distributed.

We include this in lines 172-185 in the marked-up version of revised manuscript.

Line 259: What do you mean by electricity?

**Our reply:** We are sorry for our careless mistakes and the misleading we caused. Thank you for your reminding. We restate the results in Figure 6 and include more detailed explanation of Es formation.

There are the results of spring, summer and autumn from top to bottom, respectively. Maximum Es occurrence is expected when the zonal wind shear, which is mainly produced by the semidiurnal tide in midlatitudes (Arras et al. 2009). At midlatitudes, the Es activity is dominated primarily by a semidiurnal feature, which is generally

believed to be induced by east–west zonal winds in terms of semidiurnal tides, especially in spring and summer (Whitehead 1989; Chu et al. 2014). The semidiurnal tides generally start around 6 and 14 LT, continue for 14 h, and then fade out around 20 and 4 LT separately (Tsai et al. 2018). So, it can be seen from the figure that the incidence of Es shows obvious local time changes, the period of local time 14:00-20:00 is the high incidence period of Es.
We rewrite them in lines 280-286 in the marked-up version of revised manuscript.

some minor corrections:

Line 11: As China's…
Line 19: criteria
Line 46: nineteen-thirties, … from the ground,…
Line 67: that was designed
68. concluded
73: the investigation
76: in a good agreement
80: provide very valuable
87: is introduced
93: a moving average
133: a single-layer Es event
148: shows
153: ratio
190: only by plotting
198: at a lower altitude
311: reveal
330: provided

**Our reply:** Thanks for your careful checks. We change them in the marked-up version of revised manuscript in lines 11, 19, 46, 67, 68, 73, 76, 81, 88, 94, 134, 149, 154, 205, 213, 336 and 355.

---

## Author Response (AR3)

**Response to Editor and Referees**

Dear Editor and Referees:

We appreciate very much for your kind and careful comments and useful suggestions to our manuscript, which greatly help us to improve the content and quality of manuscript. We are also very grateful to the EGU journal Annales Geophysicae for publishing our results.